# Snowflake: Scaling GNNs to high-dimensional continuous control via parameter freezing

**Charlie Blake**\*
University of Oxford
thecharlieblake@gmail.com

**Vitaly Kurin**
University of Oxford
vitaly.kurin@cs.ox.ac.uk

**Maximilian Igl**†
University of Oxford
maximilian.igl@gmail.com

**Shimon Whiteson**
University of Oxford
shimon.whiteson@cs.ox.ac.uk

## Abstract

Recent research has shown that graph neural networks (GNNs) can learn policies for locomotion control that are as effective as a typical multi-layer perceptron (MLP), with superior transfer and multi-task performance [55, 20]. However, results have so far been limited to training on small agents, with the performance of GNNs deteriorating rapidly as the number of sensors and actuators grows. A key motivation for the use of GNNs in the supervised learning setting is their applicability to large graphs, but this benefit has not yet been realised for locomotion control. We show that poor scaling in GNNs is a result of increasingly unstable policy updates, caused by overfitting in parts of the network during training. To combat this, we introduce SNOWFLAKE, a GNN training method for high-dimensional continuous control that freezes parameters in selected parts of the network. SNOWFLAKE significantly boosts the performance of GNNs for locomotion control on large agents, now matching the performance of MLPs while offering superior transfer properties.

## 1 Introduction

Whereas many traditional machine learning models operate on sequential or Euclidean (grid-like) data representations, GNNs allow for graph-structured inputs. GNNs have yielded breakthroughs in a variety of complex domains, including drug discovery [33, 50], fraud detection [56], computer vision [49, 43], and particle physics [18].

GNNs have also been successfully applied to reinforcement learning (RL), with promising results on locomotion control tasks with small state and action spaces. Not only are GNN policies as effective as MLPs on certain training tasks, but when a trained policy is transferred to another similar task, GNNs significantly outperform MLPs [55, 20]. This is largely due to the capacity of a single GNN to operate over arbitrary graph topologies (patterns of connectivity between nodes) and sizes without modification. However, so far GNNs in RL have only shown competitive performance with MLPs on lower-dimensional locomotion control tasks. For higher-dimensional tasks, one must therefore choose between superior training task performance (MLPs) and superior transfer performance (GNNs).

This paper investigates the factors underlying poor GNN scaling and introduces a method to combat them. We begin with an analysis of the GNN-based NERVENET architecture [55], which we choose for its strong zero-shot transfer performance. We show that optimisation updates for the GNN

---

\*Corresponding author. Now at Graphcore, Bristol
†Now at Waymo, Oxford

35th Conference on Neural Information Processing Systems (NeurIPS 2021).

policy have a tendency to cause excessive changes in policy space, leading to performance degrading. To combat this, current state-of-the-art algorithms [46, 48, 1] employ trust region-like constraints, inspired by natural gradients [2, 23], that limit the change in policy for each update. We outline how this policy instability can be framed as a form of overfitting—a problem GNN architectures like NERVENET are known to suffer from in supervised learning, and show that parameter regularisation (a standard remedy for overfitting) leads to a small improvement in NERVENET performance.

We then investigate which structures in the GNN contribute most to this overfitting, by applying different learning rates to different parts of the network. Surprisingly, the best performance is attained when training with a learning rate of zero in the parts of the GNN architecture that encode, decode, and propagate messages in the graph, in effect training only the part that updates node representations.

We use this approach as the basis of our method, SNOWFLAKE, which freezes the parameters of particular operations within the GNN to their initialised values, keeping them fixed throughout training while updating the non-frozen parameters as before. This simple technique enables GNN policies to be trained much more effectively in high-dimensional environments.

Experimentally, we show that applying SNOWFLAKE to NERVENET dramatically improves asymptotic performance and sample complexity on such tasks. We also demonstrate that a policy trained using SNOWFLAKE exhibits improved zero-shot transfer compared to regular NERVENET or MLPs on high-dimensional tasks.

## 2 Background

### 2.1 Reinforcement Learning

We formalise an RL problem as a Markov decision process (MDP). An MDP is a tuple $\langle \mathcal{S}, \mathcal{A}, \mathcal{R}, \mathcal{T}, \rho_0 \rangle$. The first two elements define the state space $\mathcal{S}$ and the action space $\mathcal{A}$. At every time step $t$, the agent employs a policy $\pi(a_t|s_t)$ to output a distribution over actions, selects action $a_t \sim \pi(\cdot|s_t)$, and transitions from state $s_t \in \mathcal{S}$ to $s_{t+1} \in \mathcal{S}$, as specified by the transition function $\mathcal{T}(s_{t+1}|s_t, a_t)$ which defines a probability distribution over states. For the transition, the agent gets a reward $r_t = \mathcal{R}(s_t, a_t, s_{t+1})$. The last element of an MDP specifies initial distribution over states, i.e., states an agent can be in at time step zero.

Solving an MDP means finding a policy $\pi^*$ that maximises an objective, in our case the expected discounted sum of rewards $J = \mathbb{E}_\pi \left[ \sum_{t=0}^\infty \gamma^t r_t \right]$, where $\gamma \in [0, 1)$ is a discount factor. Policy Gradients (PG) [52] find an optimal policy $\pi^*$ by doing gradient ascent on the objective: $\theta_{t+1} = \theta_t + \alpha \nabla_\theta J|_{\theta=\theta_t}$ with $\theta$ parameterising the policy.

Often, to reduce the variance of the gradient estimate, one learns a value function $V(s) = \mathbb{E}_\pi \left[ \sum_{t=0}^\infty \gamma^t r_t \mid s_0 = s \right]$, and uses it as a critic of the policy. In the resulting actor-critic method, the policy gradient takes the form: $\nabla_\theta J(\theta) = \mathbb{E}_{\pi_\theta} \left[ \sum_t A_t^{\pi_\theta} \nabla_\theta \log \pi_\theta(a_t|s_t) \right]$, where $A_t^{\pi_\theta}$ is an estimate of the advantage function $A_t^\pi = \mathbb{E}_\pi \left[ \sum_{t=0}^\infty \gamma^t r_t \mid a_t, s_t \right] - \mathbb{E}_\pi \left[ \sum_{t=0}^\infty \gamma^t r_t \mid s_t \right]$ [47].

### 2.2 Proximal Policy Optimisation

Proximal policy optimisation (PPO) [47] is an actor-critic method that has proved effective for a variety of domains including locomotion control [17]. PPO approximates the natural gradient using a first order method, which has the effect of keeping policy updates within a "trust region". This is done through the introduction of a *surrogate objective* to be optimised:

$$J = \mathbb{E}_{\pi_{\theta'}} \left[ \min \left( \frac{\pi_\theta(a|s)}{\pi_{\theta'}(a|s)} A^{\pi_{\theta'}}(s, a), \text{clip} \left( \frac{\pi_\theta(a|s)}{\pi_{\theta'}(a|s)}, 1 - \epsilon, 1 + \epsilon \right) A^{\pi_{\theta'}}(s, a) \right) \right] \tag{1}$$

where $\epsilon$ is a clipping hyperparameter that effectively limits how much a state-action pair can cause the overall policy to change at each update. This objective is computed over a number of optimisation epochs, each of which gives an update to the new policy $\pi_\theta$. If during this process a state-action pair with a positive advantage $A^{\pi_{\theta'}}(s, a)$ reaches the upper clipping boundary, the objective no longer provides an incentive for the policy to be improved with respect to that data point. This similarly applies to state-action pairs with a negative advantage if the lower clipping limit is reached.

## 2.3 Graph Neural Networks

GNNs are a class of neural architecture designed to operate over graph-structured data. We define a graph as a tuple $\mathcal{G} = (V, E)$ comprising a set of nodes $V$ and edges $E = \{(u, v) \mid u, v \in V\}$. A labelled graph has corresponding feature vectors for each node and edge that form a pair of matrices $\mathcal{L}_\mathcal{G} = (\boldsymbol{V}, \boldsymbol{E})$, where $\boldsymbol{V} = \{\mathbf{v}_v \in \mathbb{R}^p \mid v \in V\}$ and $\boldsymbol{E} = \{\mathbf{e}_{u,v} \in \mathbb{R}^q \mid (u, v) \in E\}$. For GNNs we often consider directed graphs, where the order of an edge $(u, v)$ defines $u$ as the sender and $v$ as the receiver.

A GNN takes a labelled graph $\mathcal{G}$ and outputs a second graph $\mathcal{G}'$ with new labels. Most GNN architectures retain the same topology for $\mathcal{G}'$ as used in $\mathcal{G}$, in which case a GNN can be viewed as a mapping from input labels $\mathcal{L}_\mathcal{G}$ to output labels $\mathcal{L}_{\mathcal{G}'}$.

A common GNN framework is the message passing neural network (MPNN) [14], which generates this mapping using T steps or 'layers' of computation. At each layer $\tau \in \{0, \ldots, T-1\}$ in the network, a *hidden state* $\mathbf{h}_v^{\tau+1}$ and *message* $\mathbf{m}_v^{\tau+1}$ is computed for every node $v \in V$ in the graph.

An MPNN implementation calculates these through its choice of *message functions* and *update functions*, denoted $M^\tau$ and $U^\tau$ respectively. A message function computes representations from hidden states and edge features, which are then aggregated and passed into an update function to compute new hidden states:

$$\mathbf{m}_v^{\tau+1} = \sum_{u \in N(v)} M^\tau\left(\mathbf{h}_u^\tau, \mathbf{h}_v^\tau, \mathbf{e}_{u,v}\right), \qquad \mathbf{h}_v^{\tau+1} = U^\tau\left(\mathbf{h}_v^\tau, \mathbf{m}_v^{\tau+1}\right), \tag{2}$$

for all nodes $v \in V$, where $N(v) = \{u \mid (u, v) \in E\}$ is the neighbourhood of all sender nodes connected to receiver $v$ by a directed edge. The node input labels $\mathbf{v}_v$ are used as the initial hidden states $\mathbf{h}_v^0$. MPNN assumes only *node* output labels are required, using each final hidden state $\mathbf{h}_v^T$ as the output label $\mathbf{v}_v'$.

## 2.4 NerveNet

NERVENET is an MPNN designed for locomotion control, based on the gated GNN architecture [32]. NERVENET uses the morphology (physical structure) of the agent as the basis for the GNN's input graph $\mathcal{G}$, with edges representing body parts and nodes representing the joints that connect them.

NERVENET assumes an MDP where the state $s$ can be factored into input labels $\boldsymbol{V}$, which are fed to the GNN to generate output labels: $\mathbf{V}' = \text{NERVENET}(\mathcal{G}, \mathbf{V})$. These are then used to parameterise a normal distribution defining the stochastic policy: $\pi(a|s) = \mathcal{N}(\mathbf{V}', \text{diag}(\boldsymbol{\sigma}^2))$, where the standard deviation is a separate vector of parameters learned during training. Actions $a$ are vectors, where each element represents the force to be applied at a given joint for the subsequent timestep. The policy is trained using PPO, with parameter updates computed via the Adam optimisation algorithm [25].

Internally, NERVENET uses an encoder $F_{\text{in}}$ to generate initial hidden states from input labels: $\mathbf{h}_v^0 = F_{\text{in}}(\mathbf{v}_v)$. This is followed by a message function $M^\tau$ consisting of a single MLP for all layers $\tau$ that takes as input only the state of the sender node: $\mathbf{m}_v^{\tau+1} = \sum_{u \in N(v)} \text{MLP}(\mathbf{h}_u^\tau)$. The update function $U^\tau$ is a single gated recurrent unit (GRU) [9] that maintains an internal hidden state: $\mathbf{h}_v^{\tau+1} = \text{GRU}(\mathbf{m}_v^{\tau+1} \mid \mathbf{h}_v^\tau)$. NERVENET propagates through T layers of message-passing and node-updating, before applying a decoder $F_{\text{out}}$ to turn final hidden states into scalar node output labels: $\mathbf{v}_v' = F_{\text{out}}(\mathbf{h}_v^T)$. A diagram of the NERVENET architecture can be seen in Appendix A.4, Figure 10.

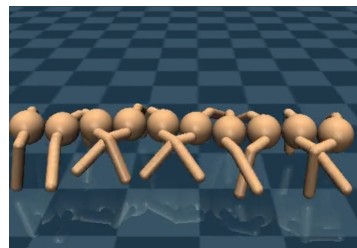 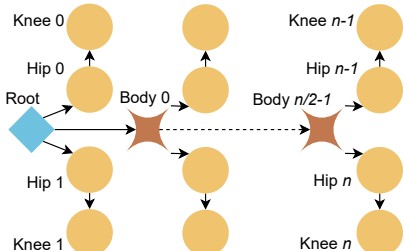

Figure 1: A MuJoCo rendering of `Centipede-20` and its corresponding morphological graph.

# 3 Analysing GNN Scaling Challenges

In this section, we use NERVENET to analyse the challenges that limit GNNs' ability to scale. We focus on NERVENET as its architecture is more closely aligned with the GNN framework than alternative approaches to structured locomotion control (see Section 4). We use mostly the same experimental setup as Wang et al. [55], with details of any differences and our choice of hyperparameters outlined in Appendix A.2.

We focus on environments derived from the Gym [8] suite, using the MuJoCo [53] physics engine. The main set of tasks we use to assess scaling is the selection of `Centipede-n` agents [55], chosen because of their relatively complex structure and ability to be scaled up to high-dimensional input-action spaces.

The morphology of a `Centipede-n` agent consists of a line of `n/2` body segments, each with a left and right leg attached (see Figure 1). The graph used as the basis for the GNN corresponds to the physical structure of the agent's body. At each timestep in the environment, the MuJoCo engine sends a feature vector containing the positions of the agent's body parts and the forces acting on them, expecting a vector to be returned specifying forces to be applied at each joint (full details of the state representation are given in Appendix A.2). The agent is rewarded for forward movement along the $y$-axis as well as a small 'survival' bonus for keeping its body within certain bounds, and given negative rewards proportional to the size of its actions and the magnitude of force it exerts on the ground.

Existing work applying GNNs to locomotion control tasks avoid training directly on larger agents, i.e., those with many nodes in the underlying graph representation. For example, Wang et al. [55] state that for NERVENET, "training a `CentipedeEight` from scratch is already very difficult". Huang et al. [20] also limit training their SMP architecture to small agent types.

## 3.1 Scaling Performance

To demonstrate the poor scaling of NERVENET to larger agents, we compare its performance on a selection of `Centipede-n` tasks to that of an MLP policy. Figure 2 shows that for the smaller `Centipede-n` agents both policies are similarly effective, but as the size of the agent increases, the performance of NERVENET drops relative to the MLP. A visual inspection of the behaviour of these agents shows that for `Centipede-20`, NERVENET barely makes forward progress at all, whereas the MLP moves effectively.

As in previous literature [e.g., 55, 20], we are ultimately not concerned with outperforming MLPs on the specific training task, but rather matching their training task performance so that the *additional* benefits of GNNs can be realised. In our setting we particularly wish to leverage the strong transfer benefits of GNNs—as demonstrated by Wang et al. [55]—resulting from their capacity to process inputs of arbitrary size and structure.

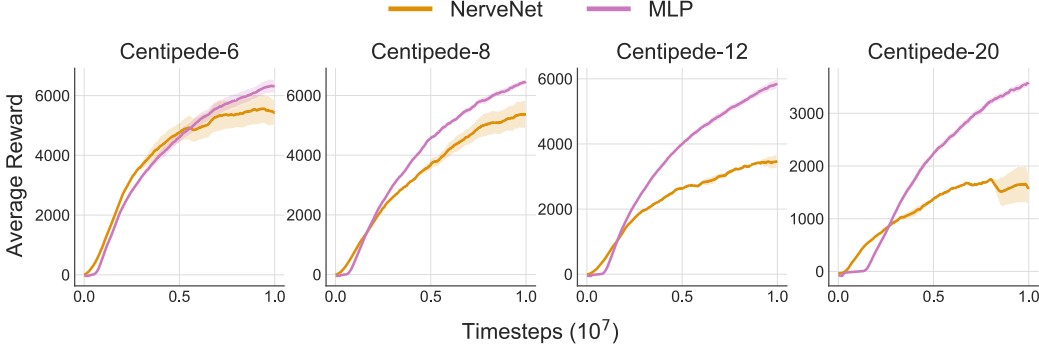

Figure 2: Comparison of the scaling of NERVENET relative to an MLP-based policy. Performance is similar for the smaller agent sizes, but NERVENET scales poorly to the larger agents.

Table 1: KL-divergence from the policy before each update to the policy after, calculated over each batch. We train on $10^7$ timesteps, recording in the table the mean taken over last $10\%$ of steps.

| Policy type | Policy KL-divergence | | | |
| --- | --- | --- | --- | --- |
| | Centipede-6 | Centipede-8 | Centipede-12 | Centipede-20 |
| MLP | 0.021 | 0.024 | 0.031 | 0.044 |
| NERVENET | 0.115 | 0.137 | 0.118 | 0.123 |

In other words, the focus of this paper is on deriving a method that can close the gap in Figure 2, as doing so makes GNNs a better choice overall given the trained policy transfers better than the MLP equivalent (see Section 5 for experimental results).

## 3.2 Unstable policy updates

As outlined in Section 2.2, one of the key challenges for on-policy RL is preventing individual updates from causing excessive changes in policy space (i.e., keeping it within the trust region). Table 1 shows the extent to which this problem contributes to NERVENET's poor scaling, calculating the average KL-divergence from the pre-update policy to the post-update policy for both policy types. NERVENET has a consistently higher KL-divergence than the MLP policy, indicating that PPO finds it harder to ensure stable policy updates for the GNN.

We emphasise that this discrepancy persists even with carefully-tuned hyperparameter values for limiting policy divergence. Figure 3 shows the performance of NERVENET across a range of PPO $\epsilon$-clipping values (see Section 2.2), and in all cases NERVENET is still substantially inferior to an MLP (note that our experiments on NERVENET always use the best value of $\epsilon = 0.1$ found here). As we demonstrate later (in Figure 8), controlling policy divergence effectively is a key component in making GNNs scale, but we see here that PPO alone does not control the divergence sufficiently to achieve this.

## 3.3 Overfitting in NERVENET

Excessive policy divergence resulting from updates can be understood as a form of overfitting. Whereas the supervised interpretation of overfitting implies poor generalisation from training to test set, in this case we are concerned with poor generalisation across state-action distributions induced

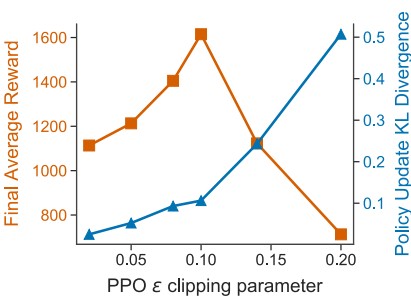

Figure 3: Final performance of NER-VENET on Centipede-20 after ten million timesteps, across a range of $\epsilon$ clipping hyperparameter values. As $\epsilon$ increases (i.e., clipping is reduced) the KL divergence from the old to new policy (blue) increases. This improves performance (orange) up to a point, after which it begins to deteriorate.

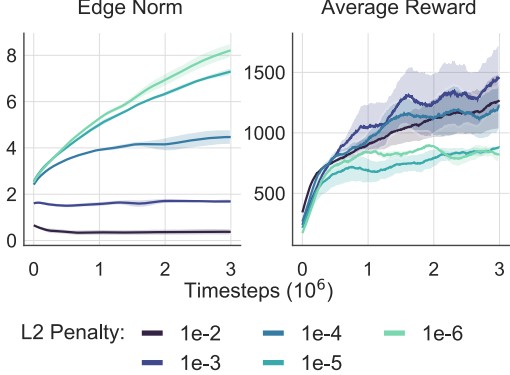

Figure 4: L2 regularisation for NERVENET's message function across a range of values for the L2 penalty $\lambda$, trained on Centipede-20. Increasing this penalty reduces the L2 norm of the weights learned (left). Improved performance for higher values of $\lambda$ (right) indicates the presence of overfitting for the message function.

by different iterations of the policy during training. Specifically, each update involves an optimisation step aiming to increase the expected reward over a batch of trajectories generated using the *pre-update* policy. The challenge for RL algorithms is that the agent is then evaluated and trained on trajectories generated using the *post-update* policy, i.e., a different distribution to the one optimised on.

For MPNN architectures like NERVENET, it is a known deficiency that in the supervised setting, message functions implemented as MLPs are prone to overfitting [16, p.55]. Here, we demonstrate that they also overfit (using the above interpretation) in our on-policy RL setting. Figure 4 shows the effect of applying L2 regularisation (a standard approach to reducing overfitting) to the NERVENET architecture. We regularise the parameters $\boldsymbol{\theta}$ of NERVENET's message function MLP $M_{\boldsymbol{\theta}}$, adding a $\lambda||\boldsymbol{\theta}||_2^2$ term to our objective function. At the optimal value of $\lambda$ we see an improvement in performance (although still substantially inferior to using an MLP), indicating that the unregularised message-passing MLPs overfit.

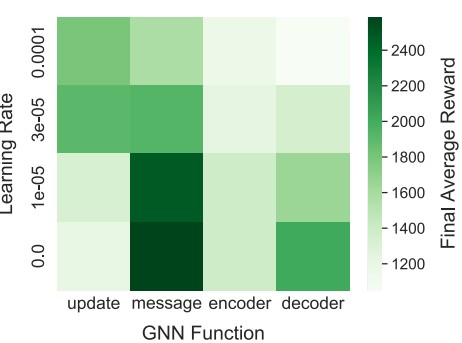

Figure 5: Colour-coded final NERVENET performance after 5M training steps on `Centipede-20` when changing learning rates for *individual* GNN components, compared to the base learning rate of $3 \times 10^{-4}$.

We also investigate lowering the learning rate in different parts of the GNN, with the aim of identifying where overfitting is localised. If parts of the network are particularly prone to damaging overfitting, training them more slowly may reduce their contribution to policy instability across updates. Results for this experiment can be seen in Figure 5.

Not only does lowering the learning rate in parts of the model improve performance, but surprisingly the best performance is obtained when the encoder $F_{\text{in}}$, message function $M$ and decoder $F_{\text{out}}$ each have their learning rate set to zero. The encoder and decoder play a similar role to the message function, all of which are implemented as MLPs, whereas the update function $U$ is a GRU (we experimented with using an MLP update function, but found that this significantly reduced performance.).

### 3.4 Snowflake

Training with a learning rate of zero is equivalent to parameter freezing (e.g., Brock et al. [7]), where parameters are fixed to their initialised values throughout training. NERVENET can learn a policy with some of its functions frozen, as learning still takes place in the un-frozen functions. For instance, if we consider freezing the encoder, this results in an arbitrary mapping of input features to the initial hidden states. As we still train the update function that processes this representation, so long as key information from the input features is not lost via the arbitrary encoding, the update function can still learn useful representations. The same logic applies to using a frozen decoder or message function.

Based on the effectiveness of parameter freezing within parts of the network, we propose a simple technique for improving the training of GNNs via gradient-based optimisation, which we name SNOWFLAKE (a naturally-occurring frozen graph structure). SNOWFLAKE assumes a GNN architecture made up internally of functions $F_{\boldsymbol{\theta}}^1, \ldots, F_{\boldsymbol{\theta}}^n$, where $\boldsymbol{\theta}$ denotes the parameters of a given function. Prior to training we select a fixed subset $\mathcal{Z} \subseteq \{F_{\boldsymbol{\theta}}^1, \ldots, F_{\boldsymbol{\theta}}^n\}$ of these functions. Their parameters are then placed in SNOWFLAKE's *frozen set* $\zeta = \{\boldsymbol{\theta} \mid F_{\boldsymbol{\theta}} \in Z\}$. During training, SNOWFLAKE excludes parameters in $\zeta$ from being updated by the optimiser, instead fixing them to whatever values the GNN architecture uses as an initialisation. Gradients still flow through these operations during backpropagation, but their parameters are not updated. In practice, we found optimal performance for $\zeta = \{F_{\text{in}}, F_{\text{out}}, M^{\tau}\}$, i.e. when freezing the encoder, decoder and message function of the GNN. If not stated otherwise, this is the architecture we refer to as SNOWFLAKE in subsequent sections. A visual representation of SNOWFLAKE applied to the NERVENET model can be seen in Figure 11, Appendix A.4.

For our experiments, we initialise the values in the GNN using the orthogonal initialisation [44]. We found this to be slightly more effective for frozen and unfrozen training than uniform and Xavier

initialisations [15]. For our message function, which has input and output dimensions of the same size, we find that performance with the frozen orthogonal initialisation is similar to that of simply using the identity function instead of an MLP. However, in the general case where the input and output dimensions of functions in the network differ (such as in the encoder and decoder, or in GNN architectures where layers use representations of different dimensionality), this simplification is not possible and freezing is required.

## 4 Related Work

**Structured Locomotion Control**    Several different graph neural network-like architectures [45, 5] have been proposed to learn policies for locomotion control. Wang et al. [55] introduce NERVENET, which trains a GNN based on the agent's morphology, along with a selection of scalable benchmarks. NERVENET achieves multi-task and transfer learning across morphologies, even in the zero-shot setting (i.e., without further training), which standard MLP-based policies fail to achieve. Sanchez-Gonzalez et al. [42] use a GNN-based architecture for learning a model of the environment, which is then used for model-predictive control.

Huang et al. [20] propose Shared Modular Policies (SMP), which focuses on multi-task training and shows strong generalisation to out-of-distribution agent morphologies using a single policy. The architecture of SMP has similarities with a GNN, but requires a tree-based description of the agent's morphology, and replaces size- and permutation-invariant aggregation with a fixed-cardinality MLP. Pathak et al. [39] propose dynamic graph networks (DGN), where a GNN is used to learn a policy enabling multiple small agents to cooperate by combining their physical structures.

Amorpheus [29] uses an architecture based on transformers [54] to represent locomotion policies. Transformers can be seen as GNNs using attention for edge-to-vertex aggregation and operating on a fully connected graph, meaning computational complexity scales quadratically with the graph size.

For all of these existing approaches to GNN-based locomotion control, training is restricted to small agents. In the case of NERVENET and DGN, emphasis is placed on the ability to perform zero-shot transfer to larger agents, but this still incurs a significant drop in performance.

**Graph-Based Reinforcement Learning**    GNNs have recently gained traction in RL due to their support for variable sized inputs and outputs, enabling new RL applications and enhancing the capabilities of agents on existing benchmarks.

Khalil et al. [24] apply DQN [38] to combinatorial optimisation problems using Structure2Vec [10] for function approximation. Lederman et al. [30] use policy gradient methods to learn heuristics of a quantified Boolean formulae solver, while Kurin et al. [28] use DQN [38] with graph networks [5] to learn the branching heuristic of a Boolean SAT solver. Klissarov and Precup [26] use a GNN to represent an MDP, which is then used to learn a form of reward shaping. Deac et al. [11] similarly use a GNN MPD representation to generalise Value Iteration Nets [51] to a broader class of MDPs.

Other approaches involve the construction of graphs based on factorisation of the environmental state into objects with associated attributes [4, 34]. In multi-agent RL, researchers have used a similar approach to model the relationship between agents, as well as between environmental objects [60, 21, 31]. In this setting, increasing the number of agents can result in additional challenges, such as combinatorial explosion of the action space. Our approach can be potentially useful to the above work, in improving scaling properties across a variety of domains.

**Random Embeddings and Parameter Freezing**    Embeddings represent feature vectors that have been projected into a new, typically lower-dimensional space that is easier for models to process. Bingham and Mannila [6] show that multiplying features by a randomly generated matrix (e.g., with entries sampled from a Gaussian distribution) preserves similarity well and empirically attains comparable performance to PCA. Wang et al. [57] use this approach to apply Bayesian Optimisation to high dimensional datasets by randomly projecting them into a smaller subspace. For natural language applications, commonly used pre-trained embeddings (e.g., word2vec [40], GloVe [37]) have been shown to offer only a small benefit over random embeddings on benchmark datasets [27, 12] and may offer no benefit on industry-scale data [3].

More generally, random embeddings can be induced by freezing typically-learned parameters within a model to fixed values throughout training. This approach has been explored for transformer

architectures, where fixed attention weights (either Gaussian-distributed [59] or hand-crafted [41]) show no significant drop in performance, and even freezing intermediate feedforward layers still enables surprisingly effective learning [35]. A similar technique can also be found in common fine-tuning methods, where parameters are pre-trained on another, possibly unsupervised objective, but frozen during training except for the final layer [e.g., 58, 19].

# 5 Experiments

We present experiments evaluating the performance of SNOWFLAKE when applied to NERVENET, and compare against regular NERVENET and MLP policies. We evaluate each model on a selection of MuJoCo tasks, including three standard tasks from the Gym suite [8] and the `Centipede-n` agents from Wang et al. [55]. Note that we do not train on even larger `Centipede-n` agents due to wall-clock simulation time becoming prohibitively large.

All training statistics are calculated as the mean across six independent runs (unless specified otherwise), with the standard error across runs indicated by the shaded areas on each graph. The average reward typically has high variance, so to smooth our results we plot the mean taken over a sliding window of 30 data points. Further experimental details are outlined in Appendix A.2.

**Scaling to High-Dimensional Tasks** Figure 6 compares the scaling properties of the regular NERVENET model with SNOWFLAKE. As the size of the agent increases, SNOWFLAKE significantly outperforms NERVENET with comparable asymptotic performance to the MLP. This indicates that SNOWFLAKE is successful in addressing the deficiencies of regular NERVENET training, and that freezing overfitting parameters is an effective training strategy in this setting. This holds true across locomotive agents with substantially different morphologies.

**Zero-shot transfer** An important motivation for improving GNN scaling is to harness their transfer capabilities on large tasks. Regular NERVENET is limited by the fact that it can only effectively train on and transfer between small agent sizes.[3] We show in Figure 7 that SNOWFLAKE attains exceptional zero-shot transfer performance across centipede sizes, surpassing alternative methods. SNOWFLAKE is the only method that can train a single policy that is effective on `Centipede-20` through to 12.

SNOWFLAKE therefore achieves our initial objective: combining the strong training task performance of an MLP and the strong transfer performance of regular NERVENET. As a consequence, SNOWFLAKE-trained GNNs offer the most promising policy representation for locomotion control tasks where transfer is a desirable property.

---

[3]We found that a regular NERVENET policy trained to achieve high training task performance on a smaller agent (e.g., `Centipede-6`) does not transfer effectively to larger ones, as reflected in [55, Figure 4].

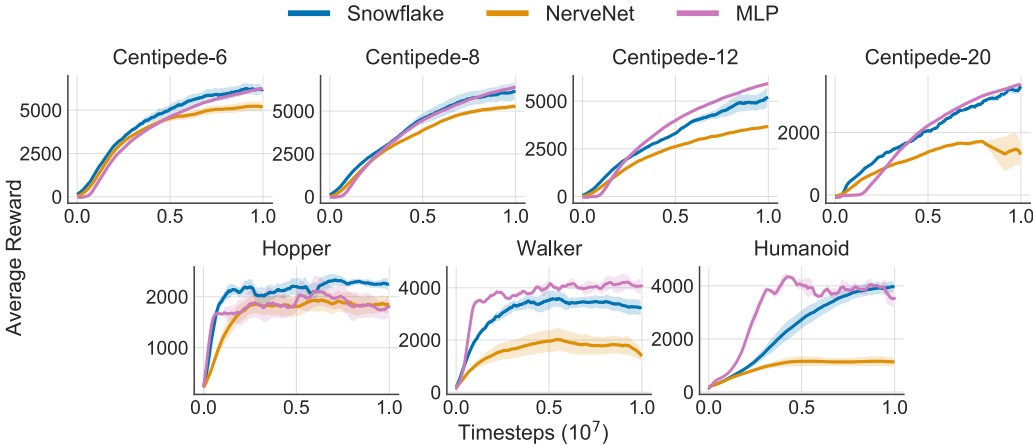

Figure 6: Comparison of the performance of SNOWFLAKE training, regular NERVENET and the MLP-based policy. SNOWFLAKE enables effective scaling to the larger agents, significantly outperforming regular NERVENET and comparable to using an MLP-based policy.

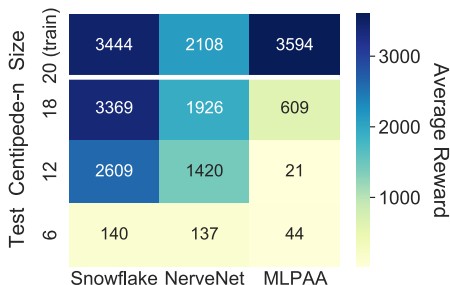

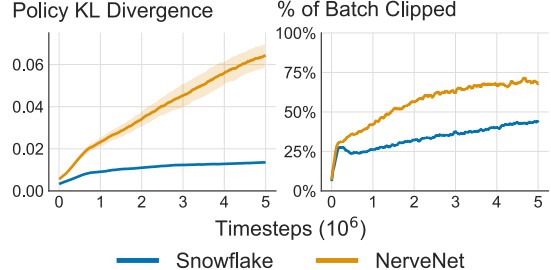

Figure 7: Zero-shot transfer performance for SNOWFLAKE, NERVENET, and MLP models trained on `Centipede-20`, evaluated across a range of sizes.

Figure 8: The effect of SNOWFLAKE on policy divergence and PPO clipping on `Centipede-20`. By freezing parts of the network that overfit, SNOWFLAKE reduces the policy KL divergence leading to less clipping during training.

**Policy Stability and Sample Efficiency**  By reducing overfitting in parts of the GNN, SNOWFLAKE mitigates the effect of harmful policy updates seen with regular NERVENET. As a consequence, the policy can train effectively on smaller batch sizes. This is demonstrated in Figure 9, which shows the performance of NERVENET trained regularly versus using SNOWFLAKE as the batch size decreases.

A potential benefit of training with smaller batch sizes is improved sample efficiency, as fewer timesteps are taken in the environment per update. However, smaller batch sizes also lead to increased policy divergence due to increased noise in the gradient estimate. When the policy divergence is too great, performance begins to decrease, limiting how small the batch can be. However due to a reduction in policy divergence as a result of SNOWFLAKE, we can afford to use smaller batch sizes while still keeping the policy under control. This provides a wider motivation for the use of SNOWFLAKE than just scaling to larger agents: it also improves sample efficiency across agents regardless of size.

The success of SNOWFLAKE in scaling to larger agents can also be understood in this context. Without SNOWFLAKE, for NERVENET to attain strong performance on large agents an infeasibly large batch size would be required, leading to poor sample efficiency. The more stable policy updates enabled by SNOWFLAKE make solving these large tasks tractable.

**PPO Clipping**  SNOWFLAKE's improved policy stability also reduces the amount of clipping performed by PPO across each training batch. Figure 8 shows the percentage of state-action pairs that are clipped for regular NERVENET versus SNOWFLAKE on the `Centipede-20` agent, as a result of reduced KL divergence[4].

When NERVENET is trained without using SNOWFLAKE a larger percentage of state-action pairs are clipped during PPO updates—a consequence of the greater policy divergence caused by overfitting. For PPO if too many data points reach the clipping limit during optimisation, the algorithm is only able to learn on a small fraction of the experience collected, reducing the effectiveness of training. One of SNOWFLAKE's strengths is that because it reduces policy divergence it requires less severe restrictions to keep the policy within the trust region. The combination of this effect and the ability to train well on smaller batch sizes enables SNOWFLAKE's strong performance on the largest agents.

## 6  Conclusion

We proposed SNOWFLAKE, a method that enables GNN-based policies to be trained effectively on much larger locomotive agents than was previously possible. We no longer observe a substantial difference in performance between using GNNs to represent a locomotion policy and the standard approach of using MLPs, even on the most challenging morphologies. As a consequence, GNNs

---

[4]It may seem counter-intuitive that the KL divergence increases over time. This is due to the standard deviation of the policy (a learned parameter) reducing as training progresses, with the agent trading exploration for exploitation.

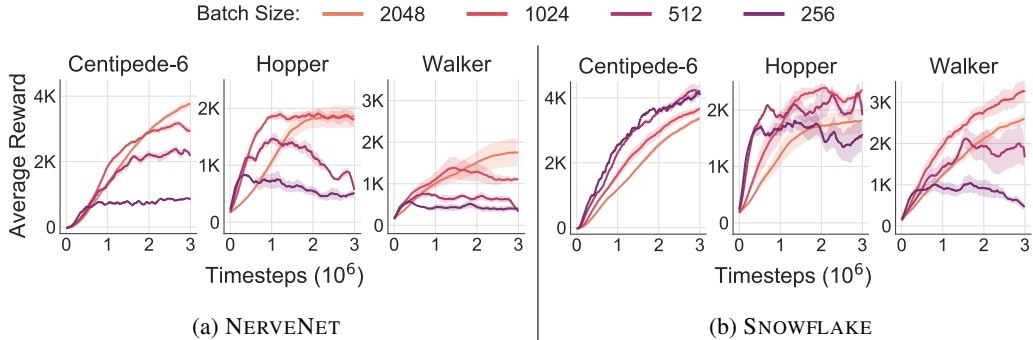

Figure 9: Effectiveness of SNOWFLAKE across smaller batch sizes relative to standard NERVENET training. SNOWFLAKE is able to use smaller batch sizes, leading to improved sample efficiency. This is due to SNOWFLAKE reducing policy divergence across updates. Corresponding policy divergence plots can be found in the appendix.

now offer an alternative to MLPs for more than just simple tasks, and if the additional features of GNNs such as strong transfer are a requirement, then they are likely to be a more effective choice. We have also provided insight into why poor scaling occurs for certain GNN architectures, and why parameter freezing is effective in addressing the overfitting problem we identify.

Limitations of our work include the upper-limit on the size of agents we were able to simulate, the use of a single algorithm and architecture, and a focus only on locomotion control tasks. Future work may include applying alternative RL algorithms and GNN architectures, schemes for automating the selection of frozen parts of the network, and applying SNOWFLAKE-like methods to a wider range of learning problems.

## Acknowledgments and Disclosure of Funding

VK is a doctoral student at the University of Oxford funded by Samsung R&D Institute UK through the AIMS program. SW has received funding from the European Research Council under the European Union's Horizon 2020 research and innovation programme (grant agreement number 637713). The experiments were made possible by a generous equipment grant from NVIDIA.

