# References

[1] Abbas Abdolmaleki, Jost Tobias Springenberg, Yuval Tassa, Rémi Munos, Nicolas Heess, and Martin A. Riedmiller. Maximum a posteriori policy optimisation. In *ICLR*, 2018.

[2] SI Amari. Neural learning in structured parameter spaces-natural riemannian gradient. *NIPS*, pages 127–133, 1997.

[3] Simran Arora, Avner May, Jian Zhang, and Christopher Ré. Contextual embeddings: When are they worth it? In *ACL*, pages 2650–2663, 2020. doi: 10.18653/v1/2020.acl-main.236.

[4] Victor Bapst, Alvaro Sanchez-Gonzalez, Carl Doersch, Kimberly Stachenfeld, Pushmeet Kohli, Peter Battaglia, and Jessica Hamrick. Structured agents for physical construction. In *ICML*, pages 464–474. PMLR, 2019.

[5] Peter Battaglia, Jessica B Hamrick, Victor Bapst, Alvaro Sanchez-Gonzalez, Vinicius Zambaldi, Mateusz Malinowski, Andrea Tacchetti, David Raposo, Adam Santoro, Ryan Faulkner, et al. Relational inductive biases, deep learning, and graph networks. *CoRR*, abs/1806.01261, 2018.

[6] Ella Bingham and Heikki Mannila. Random projection in dimensionality reduction: applications to image and text data. In *ACM*, pages 245–250. ACM, 2001. doi: 10.1145/502512.502546.

[7] Andrew Brock, Theodore Lim, James M. Ritchie, and Nick Weston. Freezeout: Accelerate training by progressively freezing layers. *CoRR*, abs/1706.04983, 2017.

[8] Greg Brockman, Vicki Cheung, Ludwig Pettersson, Jonas Schneider, John Schulman, Jie Tang, and Wojciech Zaremba. Openai gym. *CoRR*, abs/1606.01540, 2016.

[9] Kyunghyun Cho, Bart van Merrienboer, Çaglar Gülçehre, Dzmitry Bahdanau, Fethi Bougares, Holger Schwenk, and Yoshua Bengio. Learning phrase representations using RNN encoder-decoder for statistical machine translation. In *EMNLP*, pages 1724–1734. ACL, 2014. doi: 10.3115/v1/d14-1179.

[10] Hanjun Dai, Bo Dai, and Le Song. Discriminative embeddings of latent variable models for structured data. In *ICML*, volume 48 of *JMLR Workshop and Conference Proceedings*, pages 2702–2711. JMLR.org, 2016.

[11] Andreea Deac, Petar Veličković, Ognjen Milinković, Pierre-Luc Bacon, Jian Tang, and Mladen Nikolić. Xlvin: executed latent value iteration nets. In *NeurIPS*, NeurIPS Deep Reinforcement Learning Workshop, 2020.

[12] Bhuwan Dhingra, Hanxiao Liu, Ruslan Salakhutdinov, and William W. Cohen. A comparative study of word embeddings for reading comprehension. *CoRR*, abs/1703.00993, 2017.

[13] Javier García and Fernando Fernández. A comprehensive survey on safe reinforcement learning. *J. Mach. Learn. Res.*, 16:1437–1480, 2015.

[14] Justin Gilmer, Samuel S. Schoenholz, Patrick F. Riley, Oriol Vinyals, and George E. Dahl. Neural message passing for quantum chemistry. In *ICML*, volume 70, pages 1263–1272. PMLR, 2017.

[15] Xavier Glorot and Yoshua Bengio. Understanding the difficulty of training deep feedforward neural networks. In *AISTATS*, volume 9 of *JMLR Proceedings*, pages 249–256. JMLR.org, 2010.

[16] William L. Hamilton. Graph representation learning. *Synthesis Lectures on Artifical Intelligence and Machine Learning*, 14(3):p55, 2020.

[17] Nicolas Heess, Dhruva TB, Srinivasan Sriram, Jay Lemmon, Josh Merel, Greg Wayne, Yuval Tassa, Tom Erez, Ziyu Wang, SM Eslami, et al. Emergence of locomotion behaviours in rich environments. *CoRR*, abs/1707.02286, 2017.

[18] Aneesh Heintz, Vesal Razavimaleki, Javier Duarte, Gage DeZoort, Isobel Ojalvo, Savannah Thais, Markus Atkinson, Mark Neubauer, Lindsey Gray, Sergo Jindariani, et al. Accelerated charged particle tracking with graph neural networks on fpgas. *CoRR*, abs/2012.01563, 2020.

[19] Neil Houlsby, Andrei Giurgiu, Stanislaw Jastrzebski, Bruna Morrone, Quentin de Laroussilhe, Andrea Gesmundo, Mona Attariyan, and Sylvain Gelly. Parameter-efficient transfer learning for NLP. In *ICML*, volume 97 of *Proceedings of Machine Learning Research*, pages 2790–2799. PMLR, 2019.

[20] Wenlong Huang, Igor Mordatch, and Deepak Pathak. One policy to control them all: Shared modular policies for agent-agnostic control. In *International Conference on Machine Learning*, pages 4455–4464. PMLR, 2020.

[21] Shariq Iqbal, Christian A. Schröder de Witt, Bei Peng, Wendelin Boehmer, Shimon Whiteson, and Fei Sha. Randomized entity-wise factorization for multi-agent reinforcement learning. In *ICML*, pages 4596–4606, 2021.

[22] Shahin Jabbari, Matthew Joseph, Michael J. Kearns, Jamie Morgenstern, and Aaron Roth. Fairness in reinforcement learning. In *ICML*, volume 70, pages 1617–1626, 2017.

[23] Sham M. Kakade. A natural policy gradient. In *NIPS*, pages 1531–1538. MIT Press, 2001.

[24] Elias B. Khalil, Hanjun Dai, Yuyu Zhang, Bistra Dilkina, and Le Song. Learning combinatorial optimization algorithms over graphs. In *NIPS*, pages 6348–6358, 2017.

[25] Diederik P. Kingma and Jimmy Ba. Adam: A method for stochastic optimization. In *ICLR*, 2015.

[26] Martin Klissarov and Doina Precup. Reward propagation using graph convolutional networks. In *NeurIPS*, 2020.

[27] Tom Kocmi and Ondrej Bojar. An exploration of word embedding initialization in deep-learning tasks. In *ICON*, pages 56–64. NLP Association of India, 2017.

[28] Vitaly Kurin, Saad Godil, Shimon Whiteson, and Bryan Catanzaro. Can q-learning with graph networks learn a generalizable branching heuristic for a SAT solver? In *NeurIPS*, 2020.

[29] Vitaly Kurin, Maximilian Igl, Tim Rocktäschel, Wendelin Boehmer, and Shimon Whiteson. My body is a cage: the role of morphology in graph-based incompatible control. In *ICLR*, 2021.

[30] Gil Lederman, Markus N. Rabe, Sanjit Seshia, and Edward A. Lee. Learning heuristics for quantified boolean formulas through reinforcement learning. In *ICLR*, 2020.

[31] Sheng Li, Jayesh K. Gupta, Peter Morales, Ross E. Allen, and Mykel J. Kochenderfer. Deep implicit coordination graphs for multi-agent reinforcement learning. In *AAMAS*, pages 764–772. ACM, 2021.

[32] Yujia Li, Daniel Tarlow, Marc Brockschmidt, and Richard S. Zemel. Gated graph sequence neural networks. In *ICLR*, 2016.

[33] Jaechang Lim, Seongok Ryu, Kyubyong Park, Yo Joong Choe, Jiyeon Ham, and Woo Youn Kim. Predicting drug-target interaction using a novel graph neural network with 3d structure-embedded graph representation. *J. Chem. Inf. Model.*, 59(9):3981–3988, 2019. doi: 10.1021/acs.jcim.9b00387.

[34] Ricky Loynd, Roland Fernandez, Asli Çelikyilmaz, Adith Swaminathan, and Matthew J. Hausknecht. Working memory graphs. In *ICML*, pages 6404–6414, 2020.

[35] Kevin Lu, Aditya Grover, Pieter Abbeel, and Igor Mordatch. Pretrained transformers as universal computation engines. *CoRR*, abs/2103.05247, 2021.

[36] Ninareh Mehrabi, Fred Morstatter, Nripsuta Saxena, Kristina Lerman, and Aram Galstyan. A survey on bias and fairness in machine learning. *CoRR*, abs/1908.09635, 2019.

[37] Tomás Mikolov, Kai Chen, Greg Corrado, and Jeffrey Dean. Efficient estimation of word representations in vector space. In *ICLR*, 2013.

[38] Volodymyr Mnih, Koray Kavukcuoglu, David Silver, Andrei A Rusu, Joel Veness, Marc G Bellemare, Alex Graves, Martin Riedmiller, Andreas K Fidjeland, Georg Ostrovski, et al. Human-level control through deep reinforcement learning. *Nature*, 518(7540):529–533, 2015. doi: 10.1038/nature14236.

[39] Deepak Pathak, Christopher Lu, Trevor Darrell, Phillip Isola, and Alexei A. Efros. Learning to control self-assembling morphologies: A study of generalization via modularity. In *NeurIPS*, pages 2292–2302, 2019.

[40] Jeffrey Pennington, Richard Socher, and Christopher D. Manning. Glove: Global vectors for word representation. In *EMNLP*, pages 1532–1543. ACL, 2014. doi: 10.3115/v1/d14-1162.

[41] Alessandro Raganato, Yves Scherrer, and Jörg Tiedemann. Fixed encoder self-attention patterns in transformer-based machine translation. In *EMNLP*, pages 556–568. ACL, 2020. doi: 10.18653/v1/2020.findings-emnlp.49.

[42] Alvaro Sanchez-Gonzalez, Nicolas Heess, Jost Tobias Springenberg, Josh Merel, Martin A. Riedmiller, Raia Hadsell, and Peter W. Battaglia. Graph networks as learnable physics engines for inference and control. In *ICML*, pages 4467–4476, 2018.

[43] Paul-Edouard Sarlin, Daniel DeTone, Tomasz Malisiewicz, and Andrew Rabinovich. Superglue: Learning feature matching with graph neural networks. In *IEEE/CVF*, pages 4937–4946, 2020. doi: 10.1109/CVPR42600.2020.00499.

[44] Andrew M. Saxe, James L. McClelland, and Surya Ganguli. Exact solutions to the nonlinear dynamics of learning in deep linear neural networks. In *ICLR*, 2014.

[45] Franco Scarselli, Marco Gori, Ah Chung Tsoi, Markus Hagenbuchner, and Gabriele Monfardini. The graph neural network model. *IEEE Trans. Neural Networks*, 20(1):61–80, 2009. doi: 10.1109/TNN.2008.2005605.

[46] John Schulman, Sergey Levine, Pieter Abbeel, Michael I. Jordan, and Philipp Moritz. Trust region policy optimization. In *ICML*, volume 37, pages 1889–1897. JMLR, 2015.

[47] John Schulman, Philipp Moritz, Sergey Levine, Michael I. Jordan, and Pieter Abbeel. High-dimensional continuous control using generalized advantage estimation. In *ICLR*, 2016.

[48] John Schulman, Filip Wolski, Prafulla Dhariwal, Alec Radford, and Oleg Klimov. Proximal policy optimization algorithms. *CoRR*, abs/1707.06347, 2017.

[49] Yantao Shen, Hongsheng Li, Shuai Yi, Dapeng Chen, and Xiaogang Wang. Person re-identification with deep similarity-guided graph neural network. In *ECCV*, volume 11219 of *Lecture Notes in Computer Science*, pages 508–526. Springer, 2018. doi: 10.1007/978-3-030-01267-0\_30.

[50] Jonathan M Stokes, Kevin Yang, Kyle Swanson, Wengong Jin, Andres Cubillos-Ruiz, Nina M Donghia, Craig R MacNair, Shawn French, Lindsey A Carfrae, Zohar Bloom-Ackerman, et al. A deep learning approach to antibiotic discovery. *Cell*, 180(4):688–702, 2020.

[51] Aviv Tamar, Sergey Levine, Pieter Abbeel, Yi Wu, and Garrett Thomas. Value iteration networks. In *NIPS*, pages 2146–2154, 2016.

[52] Philip S. Thomas and Emma Brunskill. Policy gradient methods for reinforcement learning with function approximation and action-dependent baselines. *CoRR*, abs/1706.06643, 2017.

[53] Emanuel Todorov, Tom Erez, and Yuval Tassa. Mujoco: A physics engine for model-based control. In *IEEE/RSJ International Conference on Intelligent Robots and Systems*, pages 5026–5033. IEEE, 2012. doi: 10.1109/IROS.2012.6386109.

[54] Ashish Vaswani, Noam Shazeer, Niki Parmar, Jakob Uszkoreit, Llion Jones, Aidan N Gomez, Ł ukasz Kaiser, and Illia Polosukhin. Attention is all you need. In *NIPS*, volume 30, pages 5998–6008. Curran Associates, Inc., 2017.

[55] Tingwu Wang, Renjie Liao, Jimmy Ba, and Sanja Fidler. Nervenet: Learning structured policy with graph neural networks. In *ICLR*, 2018.

[56] Xuhong Wang, Ding Lyu, Mengjian Li, Yang Xia, Qi Yang, Xinwen Wang, Xinguang Wang, Ping Cui, Yupu Yang, Bowen Sun, and Zhenyu Guo. APAN: asynchronous propagate attention network for real-time temporal graph embedding. *CoRR*, abs/2011.11545, 2020.

[57] Ziyu Wang, Frank Hutter, Masrour Zoghi, David Matheson, and Nando de Freitas. Bayesian optimization in a billion dimensions via random embeddings. *J. Artif. Intell. Res.*, 55:361–387, 2016. doi: 10.1613/jair.4806.

[58] Jason Yosinski, Jeff Clune, Yoshua Bengio, and Hod Lipson. How transferable are features in deep neural networks? In *NIPS*, pages 3320–3328, 2014.

[59] Weiqiu You, Simeng Sun, and Mohit Iyyer. Hard-coded gaussian attention for neural machine translation. In *ACL*, pages 7689–7700, 2020. doi: 10.18653/v1/2020.acl-main.687.

[60] Vinícius Flores Zambaldi, David Raposo, Adam Santoro, Victor Bapst, Yujia Li, Igor Babuschkin, Karl Tuyls, David P. Reichert, Timothy P. Lillicrap, Edward Lockhart, Murray Shanahan, Victoria Langston, Razvan Pascanu, Matthew Botvinick, Oriol Vinyals, and Peter W. Battaglia. Deep reinforcement learning with relational inductive biases. In *ICLR*, 2019.

# A Appendix

## A.1 Ethical Discussion

Our work addresses the problem of scaling GNNs for simulated locomotion control. As our data is generated and models trained solely in a simulated physics engine, the direct ethical implications of our work are minimal. However, we identify a number of potential risks emerging from extensions and alternative applications of our work. These centre on safety and bias concerns relating to robotic control, and transferring trained policies to new tasks/agents.

**Robotic Control**   As we only ever conduct rollouts of the policy in simulation, our agent is not trained with any safety constraints in mind, which would likely be a requirement for real-world applications. Safety is particularly relevant in the wider context of our work, as the aim of scaling GNNs to more complex and capable agents potentially gives rise to increasingly unsafe behaviours and outcomes in the worst-case. Future work is required to assess if existing RL safety methods [see 13] are as effective when GNN policies are used.

Although there are many beneficial use-cases for robotic agents, there is also potential for negative social outcomes. These may be through agents that are designed directly to do harm such as autonomous weapons, or that are used in socially irresponsible applications. We encourage researchers who use our methods in the pursuit of enabling new robotics applications to give consideration to such outcomes.

**Policy Transfer**   Although effective transfer has the benefit of reducing the need for further training on the target task, the resulting policy is inevitably biased towards the original training task. Algorithmic bias has been highlighted as a key challenge in recent years for AI fairness, particularly in the supervised setting [36], but also for RL algorithms [22].

In the case of GNN policy representations, this problem can arise if the graphs (and associated labels) trained on contain harmful bias. For instance, consider a GNN policy trained on (graph-based) traffic data to optimise an RL objective, such as cumulative journey time for route planning. If the training data consists only of roads in certain geographical areas, then transferring the policy to out-of-distribution areas may lead to unsuitable actions and a disparity in outcomes. Safety constraints satisfied on the training roads (e.g. limiting the number of road accidents) may also no longer be satisfied when transferring to new areas.

**Ethical Conduct**   Our training data consists entirely of simulated physical observations from the MuJoCo environment. There is no human-generated data used in our research, nor does any of our data relate to real-world phenomena (beyond the laws of physics and design of our agents). We are therefore satisfied that our use of data is appropriate and ethical.

## A.2 Further Experimental Details

Here we outline further details of our experimental approach to supplement those given in Section 5.

**Data Generation**

As typical when training PPO on simulated environments, we train a policy by interleaving two processes: first, we perform repeated rollouts of the current policy in the environment to generate on-policy training data, and second, we optimise the policy with respect to the training data collected to generate a new policy, then repeat.

To improve wall-clock training time, for larger agents we perform rollouts in parallel over multiple CPU threads, scaling from a single thread for `Centipede-6` to five threads for `Centipede-20`. Rollouts terminate once the sum of timesteps experienced across all threads reaches the training batch size. For our experiments the main computational cost as the agent size scales is the simulator, not the training of the network. Our GNN implementation is therefore not highly optimised as this is not our bottleneck.

For optimisation we shuffle the training data randomly and split the batch into eight minibatches. We perform ten optimisation epochs over these minibatches, in the manner defined by the PPO algorithm [48] (see Section 2.2).

Each experiment is performed six times and results are averaged across runs. The exceptions to this are Figure 5 where results are an average of three runs, and the `Centipede-n` tasks in Figure 6 where results are an average of ten runs.

**Hyperparameter Search**

Our starting point for selecting hyperparameters is the hyperparameter search performed by Wang et al. [55], whose codebase ours is derived from.

To ensure that we have the best set of hyperparameters for training on large agents, we ran our own hyperparameter search on `Centipede-20` for SNOWFLAKE, as seen in Table 2.

| Hyperparameter | Values |
|---|---|
| Batch size | 512, 1024, **2048**, 4096 |
| Learning rate | 1e-4, **3e-4**, 1e-5 |
| Learning rate scheduler | adaptive, **constant** |
| $\epsilon$ clipping | 0.02, 0.05, **0.1**, 0.2 |
| GNN layers | 2, **4**, 10 |
| GRU hidden state size | **64**, 128 |
| Learned action std | shared, **separate** |

Table 2: Hyperparameter search for SNOWFLAKE on `Centipede-20`. Values in bold resulted in the best performance.

Across the range of agents tested on, we conducted a secondary search over just the batch size, learning rate and $\epsilon$ clipping value for each model. For the latter two hyperparameters, we found that the values in Table 2 did not require adjusting.

For the batch size, we used the lowest value possible until training deteriorated. Using NERVENET, a batch size of 2048 was required throughout, whereas using SNOWFLAKE a batch size of 1024 was best for `Walker`, `Centipede-20` and `Centipede-12`, 512 was best for `Centipede-8` and `Centipede-6`, and 2048 for all other agents.

Wang et al. [55] provide experimental results for the NERVENET model, which we use as a baseline for our experiments. Out of the `Centipede-n` models, they provide direct training results for `Centipede-8` (see the non-pre-trained agents in their Figure 5). Our performance results are comparable, but taken over many more timesteps. Their final MLP results appear slightly different to ours at the same point (they attain roughly 500 more reward), likely due to hyperparameter tuning for performance over a different time-frame.

They also provide performance metrics for trained `Centipede-4` and `Centipede-6` agents across the models compared (their Table 1). The results reported here are significantly less than the best performance we attain for both MLP and NERVENET on `Centipede-6`. We suspect this discrepancy is due to running for fewer timesteps in their case, but precise stopping criteria is not provided.

**Computing Infrastructure**

Our experiments were run on four different machines during the project, depending on availability. These machines use variants of the Intel Xeon E5 processor (models 2630, 2699 and 2680), containing between 44 and 88 CPU cores. As running the agent in the MuJoCo environment is CPU-intensive, we observed little decrease in training time when using a GPU; hence the experiments reported here are only run on CPUs.

Runtimes for our results vary significantly depending on the number of threads allocated and batch size used. Our standard runtime for `Centipede-6` (single thread) for ten million timesteps is around 24 hours, scaling up to 48 hours for our standard `Centipede-20` configuration (five threads). Our experiments on the default MuJoCo agents also take approximately 24 hours for a single thread.

**State Space Description**

The following is a breakdown of the information sent by the environment at each timestep to the different MuJoCo node types for the `Centipede-n` benchmark. Each different `body` and `joint` node receives its own version of this set of data:

| Node Type | Observation Type | Axis |
|---|---|---|
| body | force | x |
| | force | y |
| | force | z |
| | torque | x |
| | torque | y |
| | torque | z |
| joint | position | x |
| | velocity | x |
| root | orientation | x |
| | orientation | y |
| | orientation | z |
| | orientation | a |
| | velocity | x |
| | velocity | y |
| | velocity | z |
| | angular velocity | x |
| | angular velocity | y |
| | angular velocity | z |
| | position | z |
| | force | x |
| | force | y |
| | force | z |
| | torque | x |
| | torque | y |
| | torque | z |

Table 3: Description of the state space.

The root's z-position (height) is relative to the (global) floor of the environment. For this benchmark the joints are hinge joints, meaning that there is only one degree of freedom, and its position value reflects the joint angle (note that x-axis here refers to the joint's relative axis, not the global coordinate frame).

Our algorithm only strictly considers observations to come from joints rather than from body and root nodes. In this we follow the example set by NERVENET, which for the sake of simplicity concatenates body node observations with neighbouring joint observations, treating the resulting vector as a combined joint representation, which is then fed to the GNN.

### A.3 Sources

Our source code can be found at `https://github.com/thecharlieblake/snowflake/`, alongside documentation for building the software and its dependencies. Our code is an extension of the NERVENET codebase: `https://github.com/WilsonWangTHU/NerveNet`. This repository contains the original code/schema defining the `Centipede-n` agents.

The other standard agents are taken from the Gym [8]: `https://github.com/openai/gym`. The specific hopper, walker and humanoid versions used are `Hopper-v2`, `Walker2d-v2` and `Humanoid-v2`.

For our MLP results on the Gym agents, as state-of-the-art performance baselines have been well established in this case, we use the OpenAi Baselines codebase (`https://github.com/openai/baselines`) to generate results, to ensure the most rigorous and fair comparison possible.

The MuJoCo [53] simulator can be found at: `http://www.mujoco.org/`. Note that a paid license is required to use MuJoCo. The use of free alternatives was not viable in our case as our key benchmarks are all defined for MuJoCo.

## A.4 Supplementary Figures

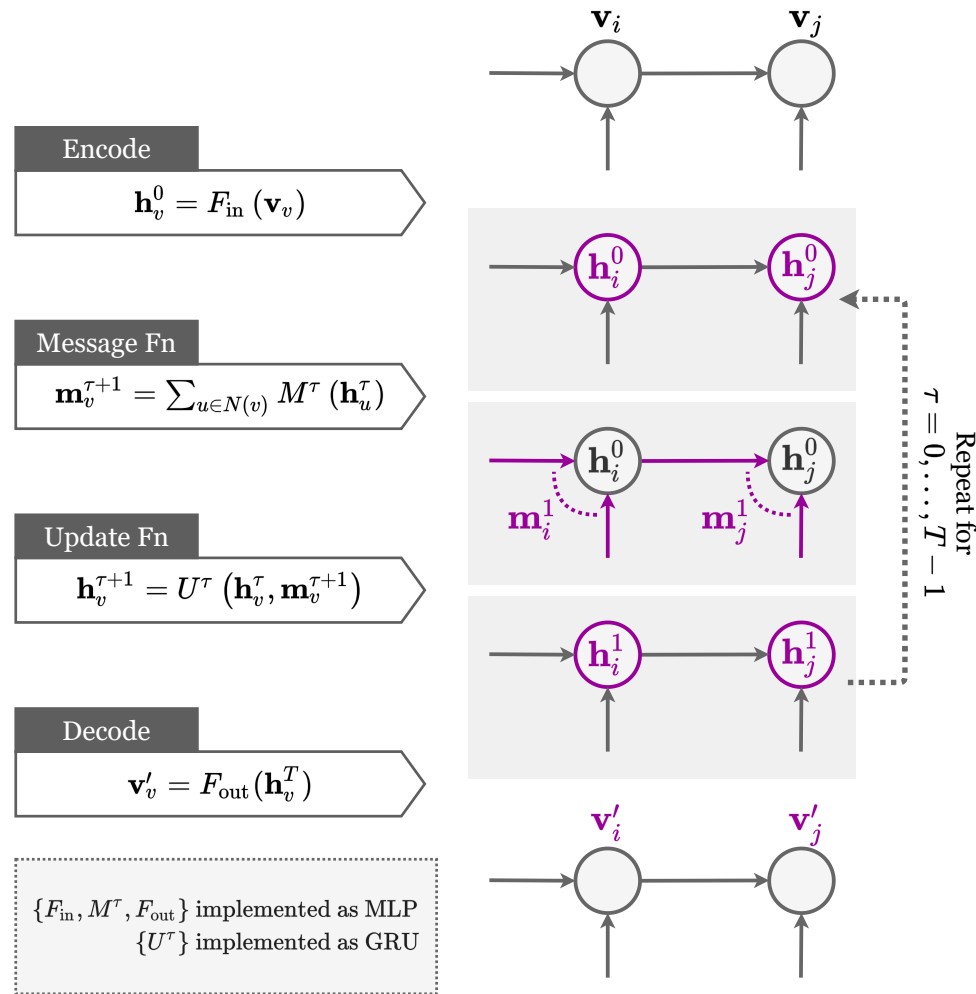

Encode
$$\mathbf{h}_v^0 = F_{\text{in}}\left(\mathbf{v}_v\right)$$

Message Fn
$$\mathbf{m}_v^{\tau+1} = \sum_{u \in N(v)} M^\tau\left(\mathbf{h}_u^\tau\right)$$

Update Fn
$$\mathbf{h}_v^{\tau+1} = U^\tau\left(\mathbf{h}_v^\tau, \mathbf{m}_v^{\tau+1}\right)$$

Decode
$$\mathbf{v}_v' = F_{\text{out}}\left(\mathbf{h}_v^T\right)$$

$\{F_{\text{in}}, M^\tau, F_{\text{out}}\}$ implemented as MLP
$\{U^\tau\}$ implemented as GRU

Repeat for $\tau = 0, \ldots, T-1$

Figure 10: A visual representation of the NERVENET architecture. Updated representations at each step are indicated in purple. Given an input vector $\mathbf{v}$ at each node, NERVENET computes scalar outputs $\mathbf{v}'$ through a series of propagation steps. Initially, the encoder is used to compute hidden states $\mathbf{h}$ at each node. These are passed into the message function, which computes an incoming message $\mathbf{m}$ for each node based on the hidden states of its neighbours. The update function then computes a new hidden state representation for each node based on the incoming message and the previous hidden state. The message function and update function then repeat their operations $T$ times, before feeding the final hidden states to the decoder, which produces outputs $\mathbf{v}'$.

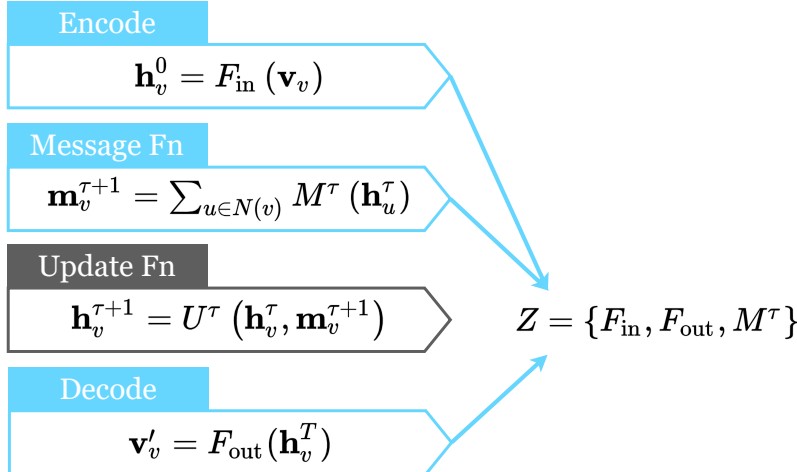

Figure 11: A visual representation of our SNOWFLAKE algorithm, as outlined in Section 3.4. Prior to training we select a fixed subset $\mathcal{Z} \subseteq \{F_{\boldsymbol{\theta}}^1, \ldots, F_{\boldsymbol{\theta}}^n\}$ of the GNN's functions. For our experiments we use $\zeta = \{F_{\text{in}}, F_{\text{out}}, M^\tau\}$. Their parameters are then placed in SNOWFLAKE's *frozen set* $\zeta = \{\boldsymbol{\theta} \mid F_{\boldsymbol{\theta}} \in Z\}$. During training, SNOWFLAKE excludes parameters in $\zeta$ from being updated by the optimiser.

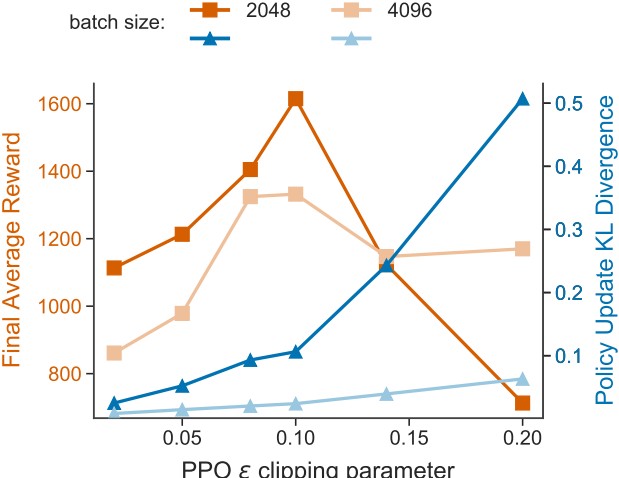

Figure 12: The effect of increasing the batch size on the influence of NERVENET's $\epsilon$ clipping hyperparameter (see Figure 3) after ten million timesteps. Increasing the batch size reduces the underlying policy divergence. This makes the algorithm less sensitive to high values of $\epsilon$ (i.e. low clipping), but also leads to a drop in sample efficiency, reducing the maximum reward attained within this time-frame.

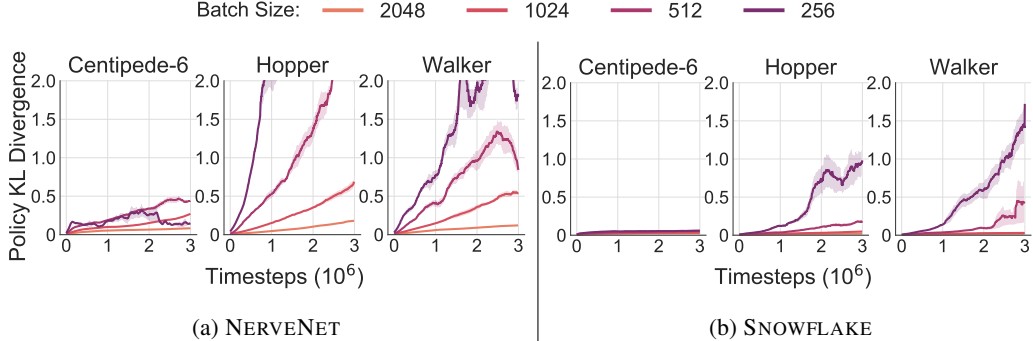

Figure 13: Accompanying KL divergence plots for Figure 9. As SNOWFLAKE reduces the policy divergence between updates, smaller batch sizes can be used before the KL divergence becomes prohibitively large. This effect underlies the improved sample efficiency demonstrated.

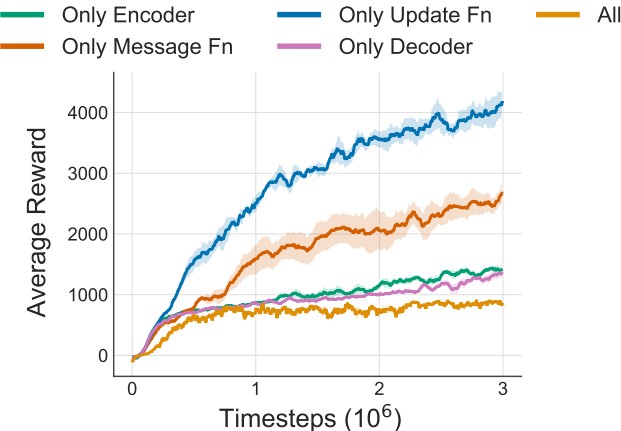

Figure 14: Ablation demonstrating the effect of only training single parts of the network (freezing the rest). The configuration of SNOWFLAKE we use for our experiments is equivalent to only training the update function, which is the most effective approach here, and all approaches are superior to training the entire GNN. For this experiment, we train on `Centipede-6` using the small batch size of 256 in all cases. This setting was chosen as it demonstrates the difference in performance for these approaches most clearly.