# OpenReview forum: "Snowflake: Scaling GNNs to high-dimensional continuous control via parameter freezing"
_NeurIPS.cc/2021/Conference — NeurIPS 2021 Poster_

### Official Review · Reviewer_X8TF · 2021-07-03

**Rating:** 6
**Confidence:** 3

**Summary:**

* The authors study a previously published method for learning locomotion in 3D agents, called NerveNet, which models the agent as a graph of joints linked by limbs and uses a Graph Neural Network (GNN) mechanism to compute joint forces. Previous work has shown that this method transfers better than MLPs across morphologies, but also scales poorly to larger agents.

* The authors identify a problem that they suggest causes this poor scaling - namely, instability of policy updates, even at performance-optimal choices of hyperparameters. They choose to call this "overfitting" - a contentious choice no doubt, but why not. They show that a L2 regularization penalty on weights alleviates the problem a bit, but not enough.

* Crucially, they also discover that reducing learning rates to zero for the encoding and decoding components of the model (i.e. using random frozen embeddings) considerably improves performance.

* Somehow they decide to give a name to the fact of using frozen random embeddings, calling it "Snowflake".

* They compare this method to standard NerveNet and MLPs on various locomotion tasks. They show that "Snowflake", unlike vanilla NerveNet, performs about as well as MLPs even on large morphologies (after enough training). Importantly, they also show that "Snowflake" trained on a given morphology transfers even better to a smaller morphology than NerveNet (and that MLPs simply can't do that at all).

* In short, the contribution of this paper (as I understand it) is an identification of a source for poor NerveNet performance, a solution to this problem (just use random frozen embeddings), and a measurement of the improved performance (matches MLPs in performance, considerably beats them in transfer).

**Limitations And Societal Impact:**

Unless I missed it (very possible), it would be useful to have a brief mention of computational costs. Is NerveNet/Snowflake slower or faster than MLPs (with similar number of trainable parameters) *per timestep*, for the particular settings used in Section 5? Even a single sentence or two would be enough.

Societal impacts do not seem applicable here.

**Main Review:**

The paper is interesting and convincing. My main question is about significance: does the contribution (improving a previously known technique) warrant a NeurIPS presentation?

I suppose it could, given the strength of the performance improvement in comparison to MLP (largely closing the gaps), the remarkable transfer results, and the possibility that this finding might apply to other GNN methods (which are evidently a hot topic right now).

Conversely, I'm a bit puzzled by the decision to coin a new name for simply using frozen random embeddings. Is it really warranted?

In the end, although I like the paper, I could see the decision go either way.

Separate from this overall uncertainty about significance, I have various clarity concerns that should be easily fixable.

1- In the (nice and useful) description of the GNN method, the notation makes it look like the inputs and outputs v_v and v'_v have the same dimensionality, and that the v' replace the v for the next step (since that's what happens between theta' and theta in the description of PPO that precedes it) ?? Surely that is not the case - it seems to me that the "input labels" v_v are fixed, unchanging descriptors of the vertices, while the "output labels" are basically just arbitrary outputs with little connection to the labels? If so this should be clarified (and perhaps the notation could be tweaked a bit).

2- Importantly, unless I missed it, the exact inputs are not clearly described.  In p.4, l. 129, we are told that they consist of "positional information" about joints and body parts. What does this information look like? The precise coding of inputs might have a great impact on performance and especially on generalization across body sizes. For example, is the "position" along the body axis fed as a simple natural integer from 0 to n/2, or is it a fractional number between 0 and 1, normalized to the total length n/2 of the body? Clearly the latter might help any architecture generalize to various lengths. Therefore, I believe a clear description of input format and information should be in the main text.

3- Relatedly, the same line mentions body parts (the edges of the network), but in section 2.4 it seems that only joint (vertices) descriptors are used? Could this be clarified?

4- Supposedly, in the experimental comparisons, MLPs have a roughly similar number of trainable parameters to NervNet/Snowflake? Can this be confirmed? In addition, it would be nice to mention how much computational cost the method incurs in comparison to MLPs (see below).



**Time Spent Reviewing:**

4

---

> ### Author Response · Authors · 2021-08-10
> **Response to Reviewer X8TF**
>
> ### Significance:
>
> On this point the reviewer largely makes our argument for us: given our very strong transfer results, closing the gap to MLPs, and the applicability to other GNN methods / settings, we think there is a compelling case that our paper's findings are significant enough to warrant publication at NeurIPS.
>
> Furthermore, we would add to that list the importance of addressing the problem of scaling. This is a key hurdle to making GNNs more widely applicable for RL, and our contribution to fixing this problem is substantial. More generally, we anticipate that our success using parameter freezing will be of interest to researchers in a broad range of areas who are looking for novel regularisation techniques.
>
> Based on the above points, along with the rest of our rebuttal, we kindly ask if the reviewer would consider increasing their score?
>
> ### Coining a name for our method:
>
> Although our method is indeed simple, we felt its novelty warranted a name, and doing so highlights that this is more than a minor training detail. In the end, we saw little harm in adding the name, and having a shorthand was convenient. We hope that the reviewer's judgement on this minor point will not be a substantial factor in their appraisal of our research.
>
> ### Question about notation:
>
> If we have understood correctly, the reviewer's feedback on our notation is that the $\theta$ and $\theta'$ symbols suggest the same dimensionality - i.e. the prime symbol indicates a new/updated version of the old parameter, whereas $v$ and $v'$ are fundamentally different variables, with different dimensionality.
> This is valuable feedback and we will update this section to use different notation for the node output labels.
>
> ### Description of inputs:
>
> We thank the reviewer for highlighting the need for this addition. We will add the following table to the appendix outlining the input observations and improve our description in the main paper. We will also alter the terminology used in the main paper, as our current description of this as 'positional information' is insufficient. Note that these benchmarks were derived not by us but by Wang et al., which we leave unaltered for the sake of comparison.
>
> The following is a breakdown of the information sent by the environment at each timestep to the different MuJoCo node types for the Centipede-n benchmark. Each different `body` and `joint` node receives its own version of this set of data:
>
> Node Type | Observation Type | Axis
> ---|---|---
> body | force | x
> body | force | y
> body | force | z
> body | torque | x
> body | torque | y
> body | torque | z
> joint | position | x
> joint | velocity | x
> root | orientation | x
> root | orientation | y
> root | orientation | z
> root | orientation | a
> root | velocity | x
> root | velocity | y
> root | velocity | z
> root | angular velocity | x
> root | angular velocity | y
> root | angular velocity | z
> root | position | z
> root | force | x
> root | force | y
> root | force | z
> root | torque | x
> root | torque | y
> root | torque | z
>
> The root's z-position (height) is relative to the (global) floor of the environment. For this benchmark the joints are hinge joints, meaning that there is only one degree of freedom, and its position value reflects the joint angle (note that x-axis here refers to the joint's relative axis, not the global coordinate frame).
>
> The reviewer is right to observe that our algorithm only strictly considers observations to come from joints rather than from body and root nodes. In this we follow the example set by NerveNet, which for the sake of simplicity concatenates body node observations with neighbouring joint observations, treating the resulting vector as a combined joint representation, which is then fed to the GNN. We will add an explanation of this to the appendix, following the above description of the inputs.
>
> ### Computational costs :
>
> For our experiments the main computational cost as the agent size scales is the simulator, not the training of the network. Our GNN implementation is slower per timestep than using an MLP, though as this was not our main bottleneck we did little to optimise it. As a consequence, a detailed comparison of the computational costs of Snowflake versus an MLP are likely to be quite specific to our implementation. However, we will add a sentence or two outlining what we have explained here to the section in our appendix where we discuss computing infrastructure.

---

### Official Review · Reviewer_soZG · 2021-07-10

**Rating:** 7
**Confidence:** 4

**Summary:**

The authors propose a framework to address the challenge of scaling the application of graph neural network (GNN) based policies to locomotion challenges in reinforcement learning. The authors' hypothesize that the scaling challenges of GNN policies relate to overfitting of internal networks within the GNN architecture during RL training and unstable policy updates during on-policy RL training using PPO. The authors then introduce their framework, named Snowflake, to address this challenge which freezes distinct parts of the GNN during training to prevent overfitting and unstable policy updates.
The authors demonstrate their framework by applying NerveNet, a GNN based policy network, to the centipede environment in the Gym + Mujoco suite. In their experiments and subsequent analysis, the authors show supporting evidence for their claim that NerveNet doesn't scale to large locomotion challenges and that the policy updates are unstable as suggested by large KL divergence between the updates when compared to regular MLP policy updates. Subsequently, the authors show how common overfitting mitigations fall short of significantly improving the performance of NerveNet in the centipede setting and provide supporting evidence that the Snowflake increases the performance of NerveNet to match regular MLP policy performance in a variety of Mujoco tasks, including various sizes in the centipede environment. The authors also claim, and support that claim, that GNN based policies, such as NerveNet, provide better generalization abilities compared to MLP based policies.

**Limitations And Societal Impact:**

The discussion on potential societal impacts, data generation and computation infrastructure in the appendix is pretty thorough. More details on limitations and potential future work of the method would make the paper stronger.

**Main Review:**

Nits:
* the checklist should start on a separate page

Overall, I thought the paper outlined their problem description, hypothesis and supporting experimental evidence pretty well. The paper provided data for their initial analysis of why GNN policies have trouble scaling to larger locomotion tasks, which is improvement based on the ICML reviews, and then show that their method improves the performance of NerveNet on various Mujoco tasks.

**Originality:** The paper proposes a novel method to address an important challenge in applying GNN based architectures to RL settings. The method can be generally applied to all GNN based architectures that are used in RL settings and potentially in other supervised learning settings.

**Quality:** The paper outlines their hypothesis and the underlying well and provides supporting evidence for their analysis of the problem, such as KL policy divergence and experiments on common regularization techniques. The experiments and results when using the method generally the method generally support the authors' claims. To further improve the paper, it would be good to clarity on a couple of items:
* How does the Snowflake choose which parts of the GNN to freeze and which parts to train? Does the method enable different learning rates across the network for the unfrozen layers (I think the authors claim it does)? If so, how are those chosen? Does applying Snowflake basically increase the space of hyperparameters in the algorithm or is there a way to systemically determine the additional choices?
* Do you expect the method to perform similarly on off-policy RL methods, such as TD3 and SAC? Why or why not?
* A discussion on the limitations of the method

**Clarity:** The paper was generally well written and provides helpful presentation of the data. Further improvements in clarity can be made by having a diagram (or algorithm block) of how the Snowflake method chooses what to freeze as also mentioned above.

**Significance:** The results in the paper indicate that Snowflake can help the adoption of GNN policies to RL problems, which would enable solving more and more complex RL challenges with greater generalization properties.

**Time Spent Reviewing:**

2

---

> ### Author Response · Authors · 2021-08-10
> **Response to Reviewer soZG**
>
> ### Choosing learning rates:
>
> Although the presentation of our algorithm is based around completely freezing/unfreezing parts of the network, from an implementation perspective it is certainly possible to apply variable learning rates to different parts of the network, which is what we do to generate the results in Figure 5.
>
> Our method for choosing which parts of the network to freeze was based around manual experimentation. In our work we simply seek to demonstrate the potential of a good freezing scheme, rather than showing how to derive an optimal one, though this is certainly valuable future work. As outlined in our paper, our process first involved a learning rate sweep across different parts of the network in isolation (Figure 5), which resulted in the conclusion that in many cases the best learning rate was zero - i.e. freezing. From that point we grouped parts that worked well frozen individually, to form a combination of frozen parts. As noted by the reviewer, considering variable learning rates does increase the hyperparameter space,  which is another reason our freezing approach is so appealing: it turns this set of continuous hyperparameters (the variable learning rates) into a set of binary ones (i.e. either freeze or use the global learning rate).
>
> ### Alternative RL algorithms:
>
> As PPO is a very widely-used algorithm—perhaps the standard choice for continuous control problems—we're satisfied that our results are valuable regardless of how many other algorithms this applies to.  We identify the problem with scaling NerveNet as being a product of overfitting in the network's message-passing architecture as the GNN is poor at generalising beyond the data it is presented with, which is likely to be an issue regardless of the algorithm used. We agree that exploring the applicability of other algorithms is worthwhile future work though. We suspect that in the off-policy setting Snowflake will give a smaller boost because stability is generally more important for on-policy training.
>
> ### Limitations and future work:
>
> At relevant points in the paper we demonstrate the limitations of particular approaches. As we outline in the checklist, we discuss some of our computational limitations in Section 5, the Humanoid results in Figure 6 reflect that Snowflake is still less sample-efficient than an MLP, and in Figure 7 we show that zero-shot transfer from Centipede-20 to Centipede-6 is still challenging for our method.
>
> We agree with the reviewer though that it would give clarity to the reader to have these observations outlined in a single part of the paper. To address this we will add a paragraph in the conclusion to outline these limitations explicitly.
> The reviewer is also right to point out that we lack a discussion of future work, which we will also add to the conclusion. This will include attempting to train GNNs for RL problems at larger scale (for instance, the GNN SAT solver by Kurin et al. 2020 had problems scaling to large graphs), applying alternative RL algorithms (e.g. TD3/SAC), exploring schemes for automating the selection of frozen parts of the network and the study of alternative GNN architectures.

---

> > ### Comment · Reviewer_soZG · 2021-08-17
> > **Thank you for the Clarifications**
> >
> > Thank you for the clarifications on my questions and comments. I was wondering if you could comment on whether you think an algorithm diagram or algorithm block to describe the full Snowflake process would be useful? If so, how do you plan to integrate it?

---

> > > ### Author Response · Authors · 2021-08-28
> > > **Diagram of the snowflake process**
> > >
> > > Based on the reviewer's feedback we have drawn up a diagram of the Snowflake process which we will add to the paper. We have designed this to complement figure 10 in the appendix (which visualises the NerveNet architecture), so we propose to add it directly after this. Doing this in the same style/format as our existing diagram we think is the clearest approach for the reader. We will also add a reference to this new diagram in section 3.4 of the paper.
> > >
> > > A link to our proposed diagram can be found here: https://anonymous.4open.science/api/repo/sf_review_diagram-5DF1/file/Screenshot%202021-08-24%20at%2019.21.54.png

---

### Official Review · Reviewer_PQSJ · 2021-07-11

**Rating:** 7
**Confidence:** 4

**Summary:**

The paper considers training RL agents by using graph neural networks (GNNs) as the function approximator, therefore inducing a useful inductive bias. Training GNNs is not without its hurdles and thus far previous work has focused on relatively small agents, putting into question scalability. This work attempts to address this issue by propposing parameter freezing in parts of the GNN. The hypothesis for this is that policy updates become unstable with GNNs, to which authors provide empirical evidence. The experiments also show better performance in more complex agents and zero-shot transfer than the NerveNet baseline.

**Limitations And Societal Impact:**

The societal impact is discussed in the appendix, but in this context is not particularly significant.

For the limitations, the authors mention that in section 5 it is discussed how their approach is hard to scale beyond centipede-20. Is it possible to provide the wall clock performance across different configurations and compare it to an MLP agent?

**Main Review:**

The intersection of GNNs and RL is an interesting and promising field of study with many applications where a graph structure can be leveraged. One such way is to consider the graph formed by the agent's anatomy, which can then be used for better transfer as shown by the authors. As GNNs become hard to train with scaling agents, the paper proposes a perhaps surprising way to fix this issue: to freeze some parts of the GNN.

The paper reads nicely and the motivation for freezing the weights is well defined, at least for the case when PPO is used. It is however not clear if this method would work or is relevant for the case then another algorithm is used, such as A2C or DDPG. What should we expect for such cases?


Another potential downside is that the transfer experiments focused exclusively on the centipede-n domain. Although it is a challenging one, it is also a limited setting. However, throughout the paper the authors refer to the "transfer properties" of GNNs, which is quite a general claim (for example in L147, where there is also no citation backing up the claim) . Such a claim would need to be backed up by experimenting on different domains, or by being more specific as to the scope of the claim.

Also, do the authors have any hypothesis as to why the KL divergence (L156) is higher when using GNNs? The idea of overfitting is an interesting one, but then again why would GNNs overfit? Perhaps L183 should be expanded a bit to give this important insight.

Figure 5 gives us a plot of the performance with respect to changes in individual parts of the GNNs (i.e. freeze one part at a time). However Snowflake freezes multiple parts of the GNNs at once. Is there any empirical evidence as to why this is done?

When freezing the message function, the authors acknowledge that this is very similar (at least in terms of performance) to using an identity function. The structure of the GNN then becomes more close to Graph Convolutional Networks [1]. Have the authors tried using GCN as a baseline?  In the same direction, it would be very informative to have an ablation or a baseline where instead of freezing the encoder and/or decoder, an identity function is used instead of a random network.

Concerning related work, the graph-based RL section could benefit from two relevant works [2,3] where GNNs are used in two quite different manners. In [2] the authors use GNNs for planning by doing more of a bread-first search approximation to the graph, where as in [3] the authors use GNNs for reward shaping by doing more of a depth-first search approximation to the graph. Adding these two references would highlight the flexibility with which GNNs can be combined with RL.

More of a minor comment, but in the background section of RL, I believe some standard references could be added as well as improving the notation (for example usually the expectation is usually explicit with respect to the policy $\pi$, as well as the value function and advantage function).

Finally, the number of runs is 6 form what I gathered in the appendix. I believe that for MuJoCo this could be easily pushed to 10 in order to provide more credibility, which is important in the current state of DRL.

[1] Kipf and Welling, Semi-Supervised Classification with Graph Convolutional Networks
[2] Deac et al., XLVIN: eXecuted Latent Value Iteration Nets
[3] Klissarov et al., Reward Propagation using Graph Convolutional Networks





=================Post-rebuttal===================

I appreciate the authors rebuttal. I think this paper is a solid contribution, and the authors rebuttal has added to this feeling. For this reason I will keep my score to accept.

**Time Spent Reviewing:**

5

---

> ### Author Response · Authors · 2021-08-10
> **Response to Reviewer PQSJ**
>
> ### Alternative RL algorithms:
>
> As PPO is a very widely-used algorithm—perhaps the standard choice for continuous control problems—we're satisfied that our results are valuable regardless of how many other algorithms this applies to.  We identify the problem with scaling NerveNet as being a product of overfitting in the network's message-passing architecture as the GNN is poor at generalising beyond the data it is presented with, which is likely to be an issue regardless of the algorithm used. We agree that exploring the applicability of other algorithms is worthwhile future work though. We suspect that in the off-policy setting Snowflake will give a smaller boost because stability is generally more important for on-policy training.
>
> ### GNN transfer properties:
>
> By 'strong transfer properties' in L147, we are referring to the ideas presented in the introduction regarding a) the previous results from the NerveNet paper showing strong GNN transfer performance on MuJoCo tasks (which we cite in L29-30), and b) the fact that GNNs can process graphs of arbitrary topology/size makes them naturally better suited to transfer than say, an MLP.
>
> We thank the reviewer for highlighting that this could be made clearer in L147, and we will amend it to make this connection more explicit.
>
> ### Why is the KL Divergence higher for GNNs?
>
> The first paragraph of 3.3 shows the connection between policy divergence and overfitting. We argue that known problems with overfitting for message-passing GNN architectures manifest themselves here in the form of increased policy divergence, and this is what's reflected in the higher KL divergence. Exactly why GNNs are afflicted by this overfitting issue is a wider problem for the field, that is beyond the scope of this paper. However, we anticipate that to those studying this problem with GNNs our work will be of great interest, for both theoreticians and practitioners.
>
> ### Why freeze multiple parts of the GNN?
>
> We would have liked to experiment with every possible freezing combination in the GNN. However, this would have required a prohibitive number of runs, particularly because multiple seeds are required to get clear results.
> Our approach was therefore to take each individual element that benefited from freezing and to combine them, resulting in the selection of frozen elements in the paper which empirically is very effective. It is an interesting direction of future work to consider more efficient or automatic schemes for identifying the best parts of a model to freeze. Here we simply seek to demonstrate the potential of a good freezing scheme, rather than showing how to derive an optimal one.
>
> ### GCN comparison:
>
> Freezing the message function certainly makes the structure of our network similar to GCN, with key difference being that we have the GRU update function, rather than the aggregation used in GCN.
>
> Experimenting further with different GNN architectures for locomotion control is an important research direction and one that our work suggests could be valuable. We chose this Gated-GNN style network because of its effectiveness in the NerveNet paper, but a more comprehensive study of the performance of different GNN architectures for locomotion control would be a valuable addition to the literature.
>
> Regarding the proposed ablation using an identity function for the encoder/decoder, this is typically not possibly as the input-output dimensionalities differ (e.g. the decoder has to reduce the node feature vectors to scalar action means).
>
> ### Additional references:
>
> We thank the reviewer for highlighting these papers and we agree that adding them to our related work would provide a broader picture of how GNNs have been applied to RL.
>
> ### Additional runs:
>
> The credibility and reproducibility of our results is something we wish to ensure. Based on the reviewer's feedback we will increase our number of runs to 10 for the experiments in figures 6 and 7 (we judge these to be the most important to prioritise our computational resources on).
>
> ### Comparing wall-clock performance:
>
> As discussed in our computing infrastructure section in the appendix, we ran our experiments across four different machines and varying numbers of parallel threads depending on compute availability. Although this did not impact the performance of our models, it did introduce significant discrepancies in wall-clock time, meaning we can't fairly compare our runs using this metric. Note that the major limitation with scaling beyond Centipede-20 is due to the simulator rather than the GNN.

---

### Official Review · Reviewer_Qzdd · 2021-07-14

**Rating:** 5
**Confidence:** 4

**Summary:**

This paper studies some of the optimization challenges with training large graph neural networks for reinforcement learning tasks, in particular locomotion control tasks. The paper proposes that one of the larger challenges in optimizing graph neural networks is the significant variance between updates across the network graph. The method proposes to randomly freeze nodes and components of nodes in the graph network. The freezing of these different blocks inside of the graph Network helps reduce the effects of large changes in the output of the network making the overall optimization less noisy and more stable resulting in better performance.

Pros
- The analysis in the paper does appear to be rather thorough. Especially going into the specifics about the difficulties in training neural networks for this particular type of problem. Also at the same time showing some of the limitations of training graph neural networks for example nerve net and its difficulty to perform well on larger agent morphologies.
- The paper then proposes a random perimeter freezing algorithm that increases the overall learning performance and final policy quality of the nerve net like design.
- Along with the performance gain the method is still able to show significant transfer capabilities across some different morphologies.

Cons
- One of the more subtle challenges in the contribution of this paper is that the method overall is somewhat simplistic. It could be thought of as performing a type of block dropout while training the network in order to reduce the amount of parameter change when making policy updates.
- One of the other more nuanced difficulties in the contribution of the paper is that it focuses on training graph neural networks for locomotion tasks on reinforcement learning. This could be seen as a somewhat specific area of research that may not have a more broad impact on the community.
- Related to the last point it would be helpful to compare this method to the amorphous work. That work calls out the challenges with training and the optimization of graph neural networks that may limit their overall application. In this new work proposed by the authors shows they've now been able to overcome some of the difficulties in training such that it may perform better than the amorphous algorithm as well.


**Limitations And Societal Impact:**

To increase the contrubution of the paper it will be important to include comparisons to other prior works, like Amorpeuos.

**Main Review:**

Additional comments on the work:
- In figure 5 it shows that setting the learning rate to practically zero for the message part of the CNN leads to the highest final average award. This also seems to be corroborated in the recent paper Amorpeuos [1], which uses Transformers to train morphological type models. Does this indicate that the training of GNNs especially the message passing system makes the overall optimization complex and could indicate that GNNs are also generally limited by the difficulty of their optimization?
- [1] This reference should be fixed in the paper to reference the proper version of the paper that has been published at ICLR 2021.
- The paper mentions the Amorpeuos prior work that uses a transformer to represent locomotion policies yet it does not compared to in the paper. In order to get a broader picture of how this method compares to the most recent state-of-the-art, it would be important to compare to this structure. Especially since one of the main points in the Amorpeuos paper was that training GNNs, especially the message passing system, is too complex in that the benefits of GNNs will be hindered by the difficulties of this optimization. However, the method proposed in this paper should be addressing these optimization limitations and therefore should perform better than the Amorpheus work.
- The analysis in figure eight with respect to how the policy KL Divergence increases over training is somewhat counterintuitive. PPO on its own does learn to adjust the policy distribution over time and as the policy should converge to at least a local optimum one would think the KL Divergence would also shrink yet in these graphs we don't see a downward trend later in learning. Does this imply that the policy is not finished learning and there is still room to learn a more optimal policy?



Refernces:
[1] Kurin, V., Igl, M., Rocktaschel, T., Boehmer, W., & Whiteson, S. (2021, April). My body is a cage: the role of morphology in graph− based incompatible control. In Proceedings of the International Conference on Learning Representations. OpenReview.

**Time Spent Reviewing:**

2.5

---

> ### Author Response · Authors · 2021-08-10
> **Response to Reviewer Qzdd**
>
> ### Comparison with Amorpheus:
>
> There were three main reasons we didn't compare Snowflake against Amorpheus:
>
> Firstly, our parameter freezing approach is orthogonal to Amorpheus' method of replacing the morphology-based graph with a fully-connected transformer. There is potentially scope to combine the two techniques. This comparison would be appropriate were the methods mutually exclusive (as is the case in our comparison with regular NerveNet). It would also naturally invite the question of what happens if we freeze parameters in Amorpheus. This is an interesting question, but examining it would require us to repeat our analysis and experiments for NerveNet entirely for Amorpheus, which is beyond the capacity of the paper.
>
> Secondly, the two methods are built around different RL algorithms. Whereas Snowflake is tested using an on-policy method (PPO), Amorpheus is tested using an off-policy method (TD3); this is done primarily because Snowflake builds off of NerveNet which also uses PPO, whereas Amorpheus builds off of SMP which uses TD3. It would not be fair to compare the two using different algorithms, and switching one approach to use the other algorithm is challenging. Not only would this require substantial implementation work given how the existing codebases are structured, but it also introduces further questions about parameter setting and design modifications.
>
> Thirdly, our primary aim was not simply to beat existing benchmarks, but rather to address a key problem in the application of GNNs to RL. There are many settings in which GNNs are better suited than transformers, and in this context comparing against a non-GNN method was less valuable to us. For instance, the scale problem we consider is one where transformers are very poorly suited, due to their quadratic complexity in the size of the graph. Even if most of the tasks we train on here are within the scope of a transformer, we anticipate that future research will scale beyond the point where fully-connected graphs are applicable, and hence solving this GNN scaling problem is important in its own right.
>
> In light of these points, as this issue was one of the reviewer's main reservations regarding our paper, we ask if they would be willing to increase their score?
>
> ### Simplicity and impact:
>
> We agree that our method is not complex, but stress that in the context of GNN training our approach is nevertheless both novel and effective. We have not seen random embeddings used as a form of regularisation elsewhere for GNNs, and as we demonstrate, on our chosen tasks it offers a large improvement. We view the simplicity of our method as one of its strengths and do not feel that this is cause for a weakness in our paper; indeed, some of the most robust ideas are the simplest ones.
>
> We do not think that using locomotion control as our specific setting entails a limited impact. Given this is a standard family of benchmarks in RL and most popular deep RL algorithms were first demonstrated on MuJoCo agents (e.g. SAC, TD3), we felt it was a natural starting-point for demonstrating our method. We anticipate that it will provide a good foundation for others to try similar approaches on more diverse RL tasks and potentially in the supervised setting.
>
> ### Block dropout:
> We are not clear on the reviewer's proposed interpretation of our algorithm as a form of block dropout. This would presumably imply that the frozen parameters aren't used at all, whereas we still use them in the forward pass, with the only difference being in the backward pass where there is no weight update.
>
> ### Why does the policy KL Divergence not decrease?
>
> This is due to the fact the standard deviations of the Gaussian from which we sample actions is also a trained parameter vector (see section 2.4). The std values are initially large, and the agent learns to gradually reduce them as it trades-off exploration for exploitation. This means that even if the policy is converging on a solution, because the peak of the action distribution reduces in width over time, the KL divergence is unlikely to decrease.
>
> We thank the reviewer for highlighting that this is not sufficiently clear, and we will amend our description of Figure 8 to explain this to the reader.

---

### Author Response · Authors · 2021-08-10
**General Response**

We would like to thank the four reviewers, both for providing valuable critical feedback, and for acknowledging the paper's merits, particularly the thoroughness of our analysis and the strength of our results.

The main reservations about our work appear to be questions about the significance of our findings (Reviewer X8TF), and the lack of a comparison against the Amorpheus model (Reviewer Qzdd). We have addressed these topics at length in our individual responses to the reviewers and hope we have satisfied them that these areas are indeed not weaknesses in our paper.

We are very grateful for the level of detail provided in the reviewers' responses, and thank them for making our research clearer to future readers.

---

### Decision · Program_Chairs · 2021-09-27

**Decision:**

Accept (Poster)

**Comment:**

There was quite an extensive discussion between the reviewers on this paper after the author response. There is no doubt that the the paper proposes a simple (in a good way) fix to train GNNs with PPO. So it makes a valuable contribution to the community. There was also agreement that the paper could potentially be made a lot stronger with additional work, e.g., the paper is mainly based on empirical evidence for the proposed method so evaluating freezing part by part (reviewer PQSJ) would be very interesting, more comparisons to other work, etc. The main difference was in the weighting of these two aspects.